# Addressing Algorithmic Disparity and Performance Inconsistency in Federated Learning

**Sen Cui[1]   Weishen Pan[1]   Jian Liang[2]   Changshui Zhang[1]   Fei Wang[3]**

[1]Institute for Artificial Intelligence, Tsinghua University (THUAI),
State Key Lab of Intelligent Technologies and Systems,
Beijing National Research Center for Information Science and Technology (BNRist),
Department of Automation,Tsinghua University, Beijing, P.R.China
[2] Alibaba Group, China
[3] Department of Population Health Sciences, Weill Cornell Medicine, USA
`{cuis19, pws15}@mails.tsinghua.edu.cn   xuelang.lj@alibaba-inc.com`
`zcs@mail.tsinghua.edu.cn   few2001@med.cornell.edu`

## Abstract

Federated learning (FL) has gain growing interests for its capability of learning from distributed data sources collectively without the need of accessing the raw data samples across different sources. So far FL research has mostly focused on improving the performance, how the algorithmic disparity will be impacted for the model learned from FL and the impact of algorithmic disparity on the utility inconsistency are largely unexplored. In this paper, we propose an FL framework to jointly consider performance consistency and algorithmic fairness across different local clients (data sources). We derive our framework from a constrained multi-objective optimization perspective, in which we learn a model satisfying fairness constraints on all clients with consistent performance. Specifically, we treat the algorithm prediction loss at each local client as an objective and maximize the worst-performing client with fairness constraints through optimizing a surrogate maximum function with all objectives involved. A gradient-based procedure is employed to achieve the Pareto optimality of this optimization problem. Theoretical analysis is provided to prove that our method can converge to a Pareto solution that achieves the min-max performance with fairness constraints on all clients. Comprehensive experiments on synthetic and real-world datasets demonstrate the superiority that our approach over baselines and its effectiveness in achieving both fairness and consistency across all local clients.

## 1   Introduction

Federated learning (FL) [1] refers to the paradigm of learning from fragmented data without sacrificing privacy. FL has aroused broad interests from diverse disciplines including high-stakes scenarios such as loan approvals, criminal justice, healthcare, etc [2]. An increasing concern is whether these FL systems induce disparity in local clients in these cases. For example, ProPublica reported that an algorithm used across the US for predicting a defendant's risk of future crime produced higher scores to African-Americans than Caucasians on average [3]. This has caused severe concerns from the public on the real deployment of data mining models and made algorithmic fairness an important research theme in recent years.

Existing works on algorithmic fairness in machine learning have mostly focused on individual learning scenarios. There has not been much research on how FL will impact the model fairness at

35th Conference on Neural Information Processing Systems (NeurIPS 2021).

different local clients[1]. Recently, Du *et al.* [6] proposed a fairness-aware method, which considered the global fairness of the model learned through a kernel re-weighting mechanism. However, such a mechanism can not guarantee to achieve fairness at local clients in FL scenario, since different clients will have different distributions across protected groups. For example, if we are building a mortality prediction model for COVID-19 patients within a hospital system [7], where each individual hospital can be viewed as a local client. Different hospitals will have different patient populations with distinct demographic compositions including race or gender. In this case, the model fairness at each hospital is important because that's where the model will be deployed, and it is unlikely that global model fairness can lead to local model fairness.

Due to the potential trade-off between algorithmic fairness and model utility, one aiming to mitigate the algorithmic disparity on local clients can exacerbate the inconsistency of the model performance (i.e., the model performance is different at different local clients). There have been researches [4, 5] trying to address the inconsistency without considering algorithmic fairness. In particular, Mohri *et al.* [5] proposed an agnostic federated learning (AFL) algorithm that maximizes the performance of the worst performing client. Li *et al.* [4] proposed a q-Fair Federated Learning (q-FFL) approach to weigh different local clients differently by taking the $q$-th power of the local empirical loss when constructing the optimization objective of the global model.

In this paper, we consider the problem of enforcing both algorithmic fairness and performance consistency across all local clients in FL. Specifically, suppose we have $N$ local clients, and $u_i$ represents the model utility for client $i$, and $g_i$ is the model disparity quantified by some computational fairness metric (e.g., demographic parity [8] or equal opportunity [9]). Following the idea of AFL, we can maximize the utility of the worst-performed client to achieve performance consistency. We also propose to assign each client a "fairness budget" to ensure certain level of local fairness, i.e., $g_i \leq \epsilon_i (\forall i = 1, 2, \cdots, N)$ with $\epsilon_i$ being a pre-specified fairness budget for client $i$. Therefore, we can formulate our problem as a constrained multi-objective optimization framework as shown in Figure 1(a), where each local model utility can be viewed as an optimization objective.

Since models with fairness and min-max performance may be not unique, we also require the model to be Pareto optimal. A model is Pareto optimal if and only if the utility of any client cannot be further optimized without degrading some others. A Pareto optimal solution of this problem cannot be achieved by existing linear scalarization methods in federated learning (e.g., federated average, or FedAve in [10]), as the non-i.i.d data distributions across different clients can cause a non-convex **Pareto Front** of utilities (all Pareto solutions form Pareto Front). Therefore, we propose FCFL, a new federated learning framework to obtain a fair and consistent model for all local clients. Specifically, we first utilize a surrogate maximum function (SMF) that considers the $N$ utilities involved simultaneously instead of the single worst, and then optimize the model to achieve Pareto optimality by controlling the gradient direction without hurting client utilities. Theoretical analysis proves that our method can converge to a fairness-constrained Pareto min-max model and experiments on both synthetic and real-world data sets show that FCFL can achieve a Pareto min-max utility distribution with fairness guarantees in each client. The source codes of FCFL are made publicly available at `https://github.com/cuis15/FCFL`.

## 2 Related Work

Algorithm fairness is defined as the disparities in algorithm decisions made across groups formed by protected variables (such as gender and race). Some approaches have been proposed to mathematically define if an algorithm is fair. For example, demographic parity [8] requires the classification results to be independent of the group memberships and equalized opportunity [9] seeks for equal false negative rates across different groups. Besides, there are also recent works focusing on the interpretation of the source of model disparity [11]. Plenty of approaches have been proposed to reduce model disparity. One type of method is to train a classifier first and then post-adjust the prediction by setting different thresholds for different groups [9] or learning a transformation function [12, 13]. Other methods have been developed for optimization of fairness metrics during the model training process through adversarial learning [14, 15, 16, 17, 18] or regularization [19, 20, 21]. Du *et al.* [6] considered algorithm fairness in the federated learning setting and proposed a regularization method that assigns

---

[1]We would like to emphasize that the algorithmic fairness we studied in this paper is not the federated fairness studied in [4] and [5], as their fairness is referring to the performance at different clients, which is consistency.

a reweighing value on each training sample for loss objective and fairness regularization to deal with the global disparity. This method cannot account for the discrepancies among the model disparities at different local clients. In this paper, we propose to treat fairness as a constraint and optimize a multi-objective optimization with multiple fairness constraints for all clients while maximally maintain the model utility.

As we introduced in the introduction, the global consensus model learned from federated learning may have different performances on different clients. There are existing research trying to address such inconsistency issue by maximizing the utility of the worst-performing client. In particular, Li *et al.* [4] propose q-FFL to obtain a min-max performance of all clients by empirically adjusting the power of the objectives, which cannot always guarantee a more consistent model utility distribution without sufficient searching for appropriate power values. Mohri *et al.* [5] propose AFL, a min-max optimization scheme which focuses on the single worst client. However, focusing on the single worst objective can cause another client to perform worse, thus we propose to take all objectives into account and optimize a surrogate maximum function to achieve a min-max performance distribution in this paper.

Multi-objective optimization aims to learn a model that gives consideration to all objectives involved. The optimization methods for multi-objective typically involve linear scalarization or its variants, such as those with adaptive weights [22], but it is challenging for these approaches to handling the competing performance among different clients [23]. Martinez *et al.* [24] proposed a multi-objective optimization framework called Min-Max Pareto Fairness (MMPF) to achieve consistency by inducing a min-max performance of all groups based on convex assumption, which is fairly strong as non-convex objectives are ubiquitous. In this paper, we formulate the problem of achieving both fairness and consistency in federated networks through constrained multi-objective optimization. Previous research on solving this problem has been mostly focusing on gradient-free algorithms such as evolutionary algorithms [25], physics-based and deterministic approaches [26]. Gradient-based methods are still under-explored [27]. We propose a novel gradient-based method FCFL , which searches for the desired gradient direction iteratively by solving constrained Linear Programming (LP) problems to achieve fairness and consistency simultaneously in federated networks.

## 3 Problem Setup

The problem to be solved in this paper is formally defined in this section. Specifically, we will introduce the algorithmic fairness problem, how to extend existing fairness criteria to federated setting, and the consistency issues of model utility in federated learning.

### 3.1 Preliminaries

**Federated Learning.** Suppose there are $N$ local clients $c_1, c_2, ...c_N$ and each client is associated with a specific dataset $\mathcal{D}^k = \left\{ X^k, Y^k \right\}, k \in \{1, 2, ..., N\}$, where the input space $X^k$ and output space $Y^k$ are shared across all $N$ clients. There are $n^k$ samples in the $k$-th client and each sample is denoted as $\left\{ x_i^k, y_i^k \right\}$. The goal of the federated learning is to collaboratively learn a global model $h$ with the parameters $\theta$ to predict the label $\hat{Y}^k$ as on each client. The classical federated learning aims to minimize the empirical risk over the samples from all clients i.e., $\min_\theta \quad \frac{1}{\sum_{k=1}^N n^k} \sum_{k=1}^N \sum_{i=1}^{n^k} l_k(h_\theta(x_i^k), y_i^k)$ where $l_k$ is the loss objective of the $k$-th client.

**Fairness.** Fairness refers to the disparities in the algorithm decisions made across different groups formed by the sensitive attribute, such as gender and race. If we denote the dataset on the $k$-th client as $D^k = \left\{ X^k, A^k, Y^k \right\}$, where $A^k \in \mathcal{A}$ is the binary sensitive attribute, then we can define the multi-client fairness as follows:

**Definition 1** (Multi-client fairness (MCF)). *A learned model $h$ achieves multi-client fairness if $h$ meets the following condition:*

$$\Delta Dis_k(h) - \epsilon_k \leq 0 \quad \forall k \in \{1, ., N\} \tag{1}$$

where $\Delta Dis_k(h)$ denotes the disparity induced by the model $h$ and $\epsilon_k$ is the given fairness budget of the $k$-th client. The disparity on the $k$-th client $\Delta Dis_k$ can be measured by *demographic parity*

(DP) [8] and *Equal Opportunity* (EO) [9] as follows:

$$\Delta DP_k = |P(\hat{Y}^k = 1 | A^k = 0) - P(\hat{Y}^k = 1 | A^k = 1)|$$
$$\Delta EO_k = |P(\hat{Y}^k = 1 | A^k = 0, Y^k = 1) - P(\hat{Y}^k = 1 | A^k = 1, Y^k = 1)| \quad (2)$$

As data heterogeneity may cause different disparities across all clients, the fairness budgets $\epsilon_k$ in Definition 3.1 specifies the tolerance of model disparity at the $k$-th client.

**Consistency.** Due to the discrepancies among data distributions across different clients, the model performance on different clients could be different. Morever, the inconsistency will be magnified when we adjust the model to be fair on local clients. There are existing research trying to improve the model consistency by maximizing the utility of the worst performing client [4, 5]:

$$\min_{\theta} \max_{k \in \{1,.,N\}} \frac{1}{n^k} \sum_{i=1}^{n^k} l_k(h_\theta(x_i^k), y_i^k)$$

where the $\max$ is over the losses across all clients.

## 3.2 Fair and Consistent Federated Learning (FCFL)

Our goal is to learn a model $h$ which 1) satisfies MCF as we defined in Definition 3.1; 2) maintains consistent performances across all clients. We will use $\Delta DP_k$ defined in Eq.(2) as measurement of disparity in our main text while the same derivations can be developed when adapting other metrics, so we have $g_k(h(X^k), A^k) = \Delta DP_k$, and $g_k$ is the function of calculating model disparity on the $k$-th client. Similarly, the model utility loss $l_k(h(X^k), Y^k)$ can be evaluated by different metrics (such as cross-entropy, hinge loss and squared loss, etc). In the rest of this paper we will use $l_k(h)$ $(g_k(h))$ for $l_k(h(X^k), Y^k)$ $(g_k(h(X^k), A^k))$ without causing further confusions. We formulate FCFLas the problem of optimizing the $N$ utility-related objectives $\{l_1(h), l_2(h), ..., l_N(h)\}$ to achieve **Pareto Min-Max** performance with $N$ fairness constraints:

$$\min_{h \in \mathcal{H}} \quad [l_1(h), l_2(h), ...l_N(h)] \quad \text{s.t. } g_k(h) - \epsilon_k \leq 0 \quad \forall k \in \{1, ., N\}. \quad (3)$$

The definitions of *Pareto Solution* and *Pareto Front*, which are fundamental concepts in multi-objective optimization, are as follows:

**Definition 2** (Pareto Solution and Pareto Front). *Suppose $l(h) = [l_1(h), l_2(h), ...l_N(h)]$ represents the utility loss vector on $N$ learning tasks with hypothesis $h \in \mathcal{H}$, we say $h$ is a Pareto Solution if there is no hypothesis $h'$ that dominates $h$: $h' \prec h$, i.e.,*

$$\nexists h' \in \mathcal{H}, \ s.t. \ \forall i : l_i(h') \leq l_i(h) \ and \ \exists j : \ l_j(h') < l_j(h).$$

*All Pareto solutions form Pareto Front $\mathcal{P}$.*

From Definition 2, for a given hypothesis set $\mathcal{H}$ and the objective vector $l(h)$, the Pareto solution avoids unnecessary harm to client utilities and may not be unique. We prefer a Pareto solution that achieves a higher consistency. Following the work in [4, 5], we want to obtain a Pareto solution $h^*$ with min-max performance. Figure 1(b) shows the relationships among different model hypothesis sets, and we explain the meanings of different notations therein as follows:

**(1)** $\mathcal{H}^F$ is the set of model hypotheses satisfying MCF, i.e.,

$$g_k(h) - \epsilon_k \leq 0 \quad \forall k \in \{1, ., N\}, \ \forall h \in \mathcal{H}^F.$$

**(2)** $\mathcal{H}^{FU}$ is the set of model hypotheses achieving min-max performance (consistency) with MCF:

$$\mathcal{H}^{FU} \subset \mathcal{H}^F \quad \text{and} \quad l_{max}(h') \leq l_{max}(h), \ \forall h \in \mathcal{H}^F, \ \forall h' \in \mathcal{H}^{FU}. \quad (4)$$

**(3)** $\mathcal{H}^{FP}$ is the set of model hypotheses achieving Pareto optimality with MCF:

$$\mathcal{H}^{FP} \subset \mathcal{H}^F \quad (5a)$$
$$h' \nprec h, \ \forall h' \in \mathcal{H}^F, \ \forall h \in \mathcal{H}^{FP} \quad (5b)$$

where Eq.(5a) satisfies $\forall h \in \mathcal{H}^{FP}$ meets MCF, and Eq.(5b) ensures that $\forall h \in \mathcal{H}^{FP}$ is a Pareto model with MCF.

**(4)** $\mathcal{H}^*$ is our desired set of model hypotheses achieving Pareto optimality and min-max performance with MCF: $\mathcal{H}^* = \mathcal{H}^{FP} \cap \mathcal{H}^{FU}$.

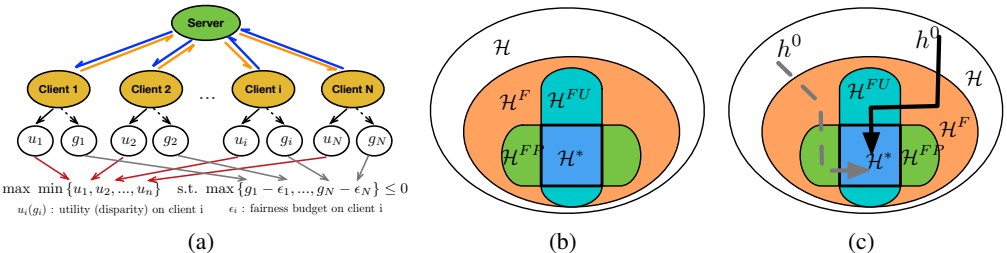

(a)                      (b)                      (c)

Figure 1: (a) The architecture of our proposed fairness-constrained min-max problem. (b) The relationship of the 5 hypothesis sets involved and $\mathcal{H}^* = \mathcal{H}^{FP} \cap \mathcal{H}^{FU}$ is the desired hypothesis set. (c) Two optimization paths to achieve a fair Pareto min-max model: 1) $h^0 \to \mathcal{H}^{FP} \to \mathcal{H}^*$: the gray dotted line represents that the initial model $h^0$ first achieves Pareto optimality with MCF then achieves min-max performance; 2) $h^0 \to \mathcal{H}^{FU} \to \mathcal{H}^*$: the black solid line denotes that the initial model $h^0$ first achieves min-max performance with MCF then achieves Pareto optimality.

**Theorem 1.** *(proof in Appendix) The hypothesis set $\mathcal{H}^*$ is non-empty, i.e.,*

$$\exists h \in \mathcal{H}, h \in \mathcal{H}^*. \tag{6}$$

In summary, from Theorem 1, our goal is to obtain a fair and consistent model $h^* \in \mathcal{H}^*$ to achieve Pareto optimality and min-max performance with MCF.

## 4 Fairness-Constrained Min-Max Pareto Optimization

### 4.1 Preliminary: Gradient-based Multi-Objective Optimization

Pareto solutions of the multi-objective optimization problem can be reached by gradient descent procedures. Specifically, given the initial hypothesis $h$ with parameter $\theta^0 \in \mathbb{R}^n$, we optimize $h_{\theta^t}$ by moving to a specific gradient direction $d$ with a step size $\eta$: $\theta^{t+1} = \theta^t + \eta d$. $d$ is a descent direction if it decreases the objectives ($l_i(h_{\theta^{t+1}}) \leq l_i(h_{\theta^t})$, $\forall i$). Suppose $\nabla_\theta l_i$ is the gradient of the $i$-th objective $l_i$ with respect to the parameter $\theta$, if we select $d^*$ which satisfies $d^{*T} \nabla_\theta l_i \leq 0$ for all $i$, $d^*$ is a descent direction and $l_i(h_{\theta^t})$ decreases after the $t+1$ iteration.

If we directly search for the descent direction $d$ to achieve $d^T \nabla_\theta l_i \leq 0, \forall i$, the computation cost can be tremendous when $d$ is a high-dimensional vector. Désidéri *et al.* [28] proposed to find a descent direction $d$ in the convex hull of the gradients of all objectives denoted as $G = [\nabla_\theta l_1, \nabla_\theta l_2, ... \nabla_\theta l_N]$ by searching for a $N$-dimension vector $\alpha^*$ (typically $N \ll n$ in deep model), which is formulated as follows:

$$d = \alpha^{*T} G \in \overline{G} \quad \text{where } \overline{G} = \left\{ \sum_{i=1}^N \alpha_i \nabla_\theta l_i \mid \alpha_j \geq 0 \; \forall j \text{ and } \sum_{j=1}^N \alpha_j = 1 \right\}. \tag{7}$$

### 4.2 Overview of the Optimization Framework

To obtain a **fair Pareto min-max** model, there are two optimization paths shown in Figure 1(c). The gray dotted line denotes the optimization path where we first achieve Pareto optimality with MCF, then we try to achieve min-max performance while keeping Pareto optimality. However, it's hard to adjust a Pareto model to another [23]. Therefore, we propose to first achieve min-max performance with MCF then achieve Pareto optimality as the black solid line in Figure 1(c). In particular, we propose a two-stage optimization method for this constrained multi-objective problem: 1) constrained min-max optimization to achieve a fair min-max model; 2) constrained Pareto optimization to continue to optimize the model to achieve Pareto optimality while keeping min-max performance with MCF.

**Constrained Min-Max Optimization** We define a constrained min-max optimization problem on the hypothesis set $\mathcal{H}$:

$$
\begin{aligned}
&\min_{h \in \mathcal{H}} \quad l_{max}(h) \\
&\qquad l_{max}(h) = \max\left(l_i(h)\right) \quad i \in \{1, ., N\} \\
&\text{s.t.} \quad g'_{max}(h) \leq 0 \\
&\qquad g'_{max}(h) = \max\left(g'_i(h)\right) = \max\left(g_i(h) - \epsilon_i\right) \quad i \in \{1, ., N\}
\end{aligned}
\tag{8}
$$

where $\epsilon_i$ is the given fairness budget on the $i$-th client. By optimizing the constrained objective in Eq.(8), we obtain a model $h^1 \in \mathcal{H}^{FU}$ that 1) $h^1$ satisfies MCF; 2) $h^1$ achieves the optimal utility on the worst performing client.

**Constrained Pareto Optimization** Caring only about the utility of the worst-performing client can lead to unnecessary harm to other clients since the rest objectives can be further optimized. Therefore, we then continue to optimize $h^1 \in \mathcal{H}^{FU}$ to achieve Pareto optimality:

$$
\min_{h \in \mathcal{H}} \frac{1}{N} \sum_{i=1}^{N} l_i(h)
\tag{9a}
$$

$$
\text{s.t. } l_i(h) \leq l_i(h^1) \quad \forall i \in \{1, ., N\}
\tag{9b}
$$

$$
g'_{max}(h) \leq 0
\tag{9c}
$$

$$
g'_{max}(h) = \max\left(g'_i(h)\right) = \max\left(g_i(h) - \epsilon_i\right) \quad i \in \{1, ., N\}
\tag{9d}
$$

where we optimize $h$ without hurting the model utility on any client so that the converged Pareto model $h^*$ of Eq.(9) satisfies $l_{max}(h^*) \leq l_{max}(h^1)$. Morever, $h^*$ satisfies MCF as the constraint in Eq.(9c), so $h^* \in \mathcal{H}^*$ is a Pareto min-max model with $N$ fairness constraints on all clients.

### 4.3 Achieving Min-Max Performance with MCF

Minimizing the current maximum value of all objectives directly in Eq.(8) can cause another worse objective and can be computationally hard when faced with a tremendous amount of clients. We will use $l_m(h)$ ($g'_m(h)$) to denote $l_{max}(h)$ ($g'_{max}(h)$) without causing further confusions for expression simplicity. We propose to use a smooth surrogate maximum function (SMF) [29] to approximate an upper bound of $l_m(h)$ and $g_m(h)$ as follows:

$$
\hat{l}_m(h, \delta_l) = \delta_l \ln \sum_{i=1}^{N} \exp\left(\frac{l_i(h)}{\delta_l}\right), \quad \hat{g}'_m(h, \delta_g) = \delta_g \ln \sum_{i=1}^{N} \exp\left(\frac{g'_i(h)}{\delta_g}\right), \quad (\delta_g, \delta_l > 0).
\tag{10}
$$

It is obvious that $l_m(h) \leq \hat{l}_m(h, \delta_l) \leq l_m(h) + \delta_m \ln(N)$ and $\lim_{\delta_l \to 0^+} \hat{l}_m(h(x), \delta_l) = l_m(h)$. For $\hat{g}'_m(h, \delta_g)$, we can get a similar conclusion. The objective in Eq.(8) is approximated as follows:

$$
\min_{h \in \mathcal{H}} \quad \hat{l}_m(h, \delta_l)
\tag{11a}
$$

$$
\text{s.t.} \quad \hat{g}'_m(h, \delta_g) \leq 0.
\tag{11b}
$$

**Property 1.** *There always exists an initial trivial model which satisfies the MCF criterion by treating all samples equally (e.g., $h(x) = 1, \forall x$).*

From Property 1, we can always initialize $h$ to satisfy $\hat{g}'_m(h, \delta_g) \leq 0$ in Eq.(11b). Then we optimize the upper bound of $l_m(h)$ when ensuring MCF. As the hypothesis $h$ owns the parameter $\theta$, we use $\nabla_\theta \hat{l}$ and $\nabla_\theta \hat{g}'$ to represent the gradient $\nabla_\theta \hat{l}_m(h_\theta, \delta_l)$ and $\nabla_\theta \hat{g}'_m(h_\theta, \delta_g)$, respectively. We propose to search for a descent direction $d$ for Eq.(11) in the convex hull of $G = [\nabla_\theta \hat{l}, \nabla_\theta \hat{g}']$ in two cases where $d$ is defined in Eq.(7). For the $t$-th iteration:

**(1)** if $h_{\theta^t}$ satisfies the fairness constraint defined in Eq.(11b), we focus on the descent of $\hat{l}_m(h_{\theta^t}, \delta_l^t)$:

$$
\min_{d \in \overline{G}} \quad d^T \nabla_{\theta^t} \hat{l} \quad \text{if } \hat{g}'_m(h_{\theta^t}, \delta_g^t) \leq 0,
\tag{12}
$$

**(2)** if $h_{\theta^t}$ violates the fairness constraint, we aim to the descent of $\hat{g}'_m(h_{\theta^t}, \delta_g^t)$ without causing the ascent of $\hat{l}_m(h_{\theta^t}, \delta_l^t)$ :

$$\min_{d \in \overline{G}} \quad d^T \nabla_{\theta^t} \hat{g}', \quad \text{s.t. } d^T \nabla_{\theta^t} \hat{l} \leq 0. \tag{13a}$$

If the obtained gradient $d$ satisfies $||d||^2 \leq \epsilon_d$, we decrease the parameter $\delta_l^t, \delta_g^t$ as:

$$\begin{aligned}
\delta_l^{t+1} &= \beta \cdot \delta_l^t, \ \delta_g^{t+1} = \beta \cdot \delta_g^t \qquad \text{if } ||d||^2 \leq \epsilon_d \\
\delta_l^{t+1} &= \delta_l^t, \ \delta_g^{t+1} = \delta_g^t \qquad\qquad \text{otherwise,}
\end{aligned} \tag{14}$$

where $\beta$ is the decay ratio that $0 < \beta < 1$. From Eq.(14), we narrow the gap between $\hat{l}_m(h, \delta_l)$ and $l_{max}(h)$ by decreasing the parameter $\delta_l$ as every time the algorithm approaches convergence. From Eq.(12) and Eq.(13), we optimize either $\hat{l}_m$ or $\hat{g}'_m$ and keep $\hat{l}_m$ without ascent in each iteration.

### 4.4 Achieving Pareto Optimality *and* Min-Max Performance with MCF

As the model $h_\theta^1 \in \mathcal{H}^{FU}$ obtained from constrained min-max optimization cannot guarantee Pareto optimality, we continue to optimize $h_\theta^1$ to be a Pareto model without causing the utility descent on any client. We propose a constrained linear scalarization objective to reach a Pareto model and the $t$-th iteration is formulated as follows:

$$\min_{d \in \overline{G}} \frac{1}{N} \sum_{i=1}^{N} d^T \nabla_{\theta^t} l_i \tag{15a}$$

$$\text{s.t.} \quad d^T \nabla_{\theta^t} l_i \leq 0 \quad \forall i \in \{1, ., N\} \tag{15b}$$

$$d^T \nabla_{\theta^t} \hat{g}'_m \leq 0, \tag{15c}$$

where $G = [\nabla_\theta l_1, \nabla_\theta l_2, ... \nabla_\theta l_N, \nabla_\theta g_1, \nabla_\theta g_2, ..., \nabla_\theta g_N]$ and $\overline{G}$ is the convex hull of $G$. The non-positive angle of $d$ with each gradient $\nabla_{\theta^t} l_i$ ensures that all objective values decrease. Similarly, if we aim to reach the Pareto solution without causing utility descent only on the worst performing client, the constraint in Eq.(15b) is replaced by $d^T \nabla_{\theta^t} \hat{l} \leq 0$.

Different from constrained min-max optimization in Section 4.3 where the objective to be optimized in each iteration depends on whether $\hat{g}_m(h_{\theta^t}, \delta_g^t) \leq 0$, in constrained Pareto optimization procedure, as we have achieved MCF, we optimize the objective in Eq.(15) for a dominate model in each iteration. Specifically, we restrict $\hat{g}'_m \leq 0$ to keep fairness on each client given the reached hypothesis $h^1 \in \mathcal{H}^{FU}$. Meanwhile, we constrain $l_i(h)$ to descend or remain unchanged until any objective cannot be further minimized. Algorithm 1 in Appendix shows all steps of our method. The convergence analysis and the time complexity analysis of our framework are in Appendix. Moreover, we also present our discussion about the fairness budget $\epsilon$ in Appendix.

## 5 Experiments

We intuitively demonstrate the behavior of our method by conducting experiments on synthetic data. For the experiments on two real-world federated datasets with fairness issues, we select two different settings to verify the effectiveness of our method: (1) assign equal fairness budgets for all local clients; (2) assign client-specific fairness budgets. More experimental results and detailed implementation are in Appendix.

### 5.1 Experimental Setup

**Federated Datasets** (1) Synthetic dataset: following the setting in [30, 23], the synthetic data is from two given non-convex objectives; (2) UCI *Adult* dataset [5]: *Adult* contains more than 40000 adult records and the task is to predict whether an individual earns more than 50K/year given other features. Following the federated setting in [4, 5], we split the dataset into two clients. One is PhD client in which all individuals are PhDs and the other is non-PhD client. In our experiments, we select *race* and *gender* as sensitive attributes, respectively. (3)*eICU* dataset: We select [31], a clinical dataset collecting patients about their admissions to ICUs with hospital information. Each instance

is a specific ICU stay. We follow the data preprocessing procedure in [32] and naturally treat 11 hospitals as 11 local clients in federated networks. We conduct the task of predicting the prolonged length of stay (whether the ICU stay is longer than 1 week, ) and select *race* as the sensitive attribute.

**Evaluation Metrics** (1) Utility metric: we use *accuracy* to measure the model utility in our experiments; (2) Disparity metrics: our method is compatible with various of fairness metrics. In our experiments, we select two metrics (marginal-based metric *Demographic Parity* [8] and conditional-based metric *Equal Opportunity* [9] (The results of *Equal Opportunity* are in the Appendix) to measure the disparities defined in Eq.(2); (3) Consistency: following the work [4, 5], we use the utility on the worst-performing client to measure consistency.

**Baselines** As we do not find prior works proposed for achieving fairness in each client, we select FA [6], MMPF [24] and build FedAve+FairReg as baselines in our experiments. For all baselines, we try to train the models to achieve the optimal utility with fairness constraints. If the model cannot satisfy the fairness constraints, we keep the minimum of disparities with reasonable utilities. (1) MMPF [24], Martinez *et al.* develop MMPF which optimizes all objectives on convex assumption to induce a min-max utility of all groups; (2) FA [6], Du *et al.* propose FA, a kernel-based model-agnostic method with regularizations for addressing fairness problem on a new unknown client instead of all involved clients; (3) FedAve+FairReg, we build FedAve+FairReg, which optimizes the linear scalarized objective with the fairness regularizations of all clients.

## 5.2 Experiments on Synthetic Dataset

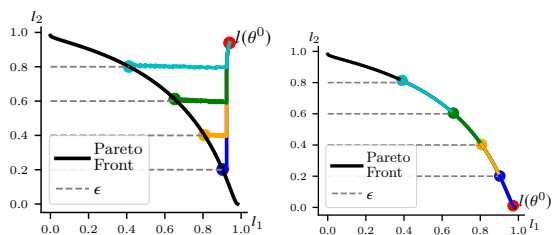

(a) $\theta^0$ violates the constraints (b) $\theta^0$ satisfies the constraints

Figure 2: Optimization trajectories of FCFL in $n = 20$ dimensional solution space ($\theta \in \mathbb{R}^{20}$). The initialization violates fairness constrains (left) and satisfies fairness constraints (right).

Following the setting in [30, 23], the synthetic data is from the two non-convex objectives to be minimized in Eq.(16) and the Pareto Front of the two objectives is also non-convex.

$$l_1(\theta) = 1 - e^{-\left\|\theta - \frac{1}{\sqrt{n}}\right\|_2^2}$$
$$l_2(\theta) = 1 - e^{-\left\|\theta + \frac{1}{\sqrt{n}}\right\|_2^2}.$$
(16)

Non-convex Pareto Front means that linear scalarization methods (e.g., FedAve) miss any solution in the concave part of the Pareto Front. In this experiment, we optimize $l_1$ under the constraint $l_2 \leq$ $\epsilon, \epsilon \in \{0.2, 0.4, 0.6, 0.8\}$. Considering the effect of the initialization in our experiment, we conduct experiments when the initialization $\theta^0$ satisfies the constraints and violates the constraints.

From the results in Figure 2, when the initialization $\theta_0$ violates the constraints in Figure 2(a), the objective $l_2$ decreases in each step until it satisfies the constraint and finally FCFL reaches the constrained optimal $l_1(h^*)$. As the initialization $\theta_0$ satisfies the constraints in Figure 2(b), our method focuses on optimizing $l_1$ until it achieves the optimal $l_1(h^*)$ with the constraint $l_2 \leq \epsilon$.

## 5.3 Experiments on Real-world Datasets with Equal Fairness Budgets

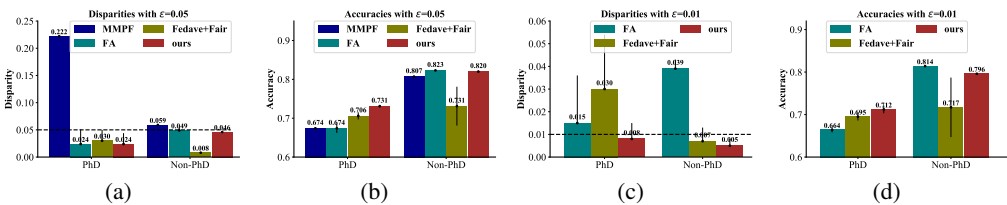

Figure 3: The disparities and accuracies on both clients as $\epsilon = 0.05$ (top) and as $\epsilon = 0.01$ of on Adult dataset when *race* is the sensitive attribute.

### 5.3.1 Income Prediction on Adult Dataset

We show the results with the sensitive attribute being *race* in our main text and the results when *gender* is the sensitive attribute are in Appendix. We set the fairness budgets defined in Eq.(1) in two different cases, (1) looser constraint: $\Delta DP_i \leq \epsilon_i = 0.05 \ \forall i \in \{1, ., N\}$; (2) stricter constraint: $\Delta DP_i \leq \epsilon_i = 0.01 \ \forall i \in \{1, ., N\}$.

From Figure 3, FCFL achieves min-max performance on PhD client in the two cases with MCF. MMPF fails to achieve MCF as $\epsilon_i = 0.05$. FA and FedAve+FairReg achieve MCF with $\epsilon_i = 0.05$ but violate fairness constraint as $\epsilon_i = 0.01$. From Figure 3, our method achieves a comparable performance on non-PhD client compared to baselines.

### 5.3.2 Prolonged Length of Stay Prediction on eICU Dataset

The length of the patient's hospitalization is critical for the arrangement of the medical institutions as it is related to the allocation of limited ICU resources. We conduct experiments on the prediction of prolonged length of stay(whether the ICU stay is longer than 1 week) on eICU dataset. We use *race* as sensitive attribute and set the fairness budgets defined in Eq.(1) in two cases: (1) looser constraint: $\Delta DP_i \leq \epsilon_i = 0.1 \ \forall i \in \{1, ., N\}$; (2) stricter constraint: $\Delta DP_i \leq \epsilon_i = 0.05 \ \forall i \in \{1, ., N\}$

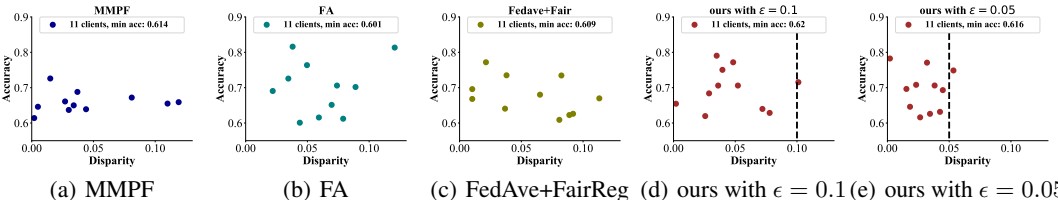

    (a) MMPF      (b) FA      (c) FedAve+FairReg  (d) ours with $\epsilon = 0.1$(e) ours with $\epsilon = 0.05$

Figure 4: Experiments on LoS Prediction task with the sensitive attribute being *race*. The points in each figure denote the clients and X and Y coordinates of the points denote the disparities and the accuracies, respectively.

From Figure 4, our method achieves min-max performance with fairness budget $\epsilon = 0.1$ compared to the baselines. When we constrain $\Delta DP_i \leq 0.05 \ \forall i$, all baselines fail to satisfy the constraints and the disparities are about 0.1 while our method significantly decreases the disparities and the maximum of the disparities is 0.68. Besides, we maintain comparable utilities on all clients compared to baselines.

### 5.4 Experiments with Client-Specific Fairness Budgets

Data heterogeneity encourages the different levels of the disparities on different clients. Consistent fairness budgets can cause unexpected hurt on some specific clients with severe disparities. We explore the performance of our method given client-specific fairness budgets. Specifically, we firstly conduct unconstrained min-max experiments and measure the original disparities $\Delta \boldsymbol{DP} = [\Delta DP_1, ., \Delta DP_N]$ on all clients, then we constrain the model disparities based on the original disparities of all clients, i.e., $[\epsilon_1, \epsilon_2, ., \epsilon_N] = w \cdot \Delta \boldsymbol{DP}, w \in \{1.0, 0.9, 0.8, ., 0.2\}$.

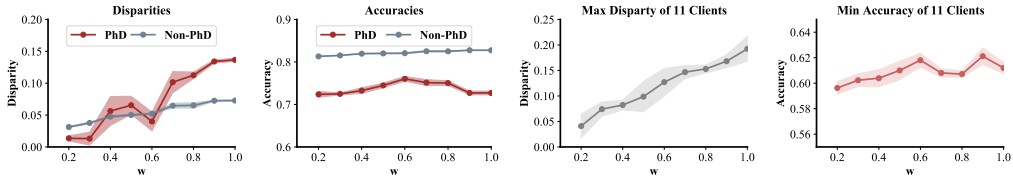

(a) *Adult Disparity* $-$ *w* (b) *Adult Accuracy* $-$ *w* (c) *eICU Disparity* $-$ *w* (d) *eICU Accuracy* $-$ *w*

Figure 5: Client-specific constraint experiment on Adult and eICU.

From Figure 5(a) and Figure 5(b), we show the effect of the client-specific fairness budgets on model disparities and utilities. The decreasing $w$ means the stricter constraints on both clients and FCFL reduces the disparities significantly as shown in Figure 5(a). With the stricter client-specific constraints on both clients, the utilities on both clients decrease slightly as shown in Figure 5(b),

which implies that FCFL achieves a great balance between the fairness and the utility of all clients. FCFL is compatible with client-specific fairness budgets which enhances its flexibility and avoids severe hurt to the specific clients.

The results of LoS prediction task with client-specific fairness budgets are shown in Figure 5(c) and Figure 5(d). As $w$ decreases from 1.0 to 0.2, the maximum of all client disparities in Figure 5(c) decrease from 0.2 to 0.05 which means the model becomes fairer on all clients. Figure 5(d) shows the minimum of the utilities which slightly decreases from 0.62 to 0.6 and our method achieves an acceptable trade-off between model fairness and utility as the amount of clients increases in this task.

## 6 Conclusion

In this paper, we investigate the consistency and fairness issues in federated networks as the learned model deployed on local clients can cause inconsistent performances and disparities without elaborate design. We propose a novel method called FCFL to overcome the disparity and inconsistency concerns in the favored direction of gradient-based constrained multi-objective optimization. Comprehensive empirical evaluation results measured by quantitative metrics demonstrate the effectiveness, superiority, and reliability of our proposed method.

## Acknowledgments

Sen Cui, Weishen Pan and Changshui Zhang would like to acknowledge the funding by the National Key Research and Development Program of China (No. 2018AAA0100701). Fei Wang acknowledges the support from AWS Machine Learning for Research Award and Google Faculty Research Award.

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
