# A  Algorithm Details

We summarize our method in Algorithm 1 as follows:

---

**Algorithm 1:** FCFL : obtain a **fair Pareto min-max** model

---

**Input:** fairness budget $\{\epsilon_i\}$, gradient threshold $\epsilon_d$, the initial parameters $\delta_l^0, \delta_g^0$, the learning rate $\eta$, the parameter decay rate $\beta$, the initial hypothesis $h_{\theta^0}$, and the data of all clients $\{D^1, D^2, ...D^N\}$.

**Stage 1:** obtain a fair min-max model $h^1$

**for** $t = 1, 2, ...T^1$ **do**

> *Local clients:*
> The center server sends the global model $h^t$ to all local client;
> Local clients evaluate the performance of $h^t$ and obtain the local gradients $\nabla_{\theta_t} l_i$;
> *Center server:*
> Calculate the surrogate maximum objective $\hat{l}$ and disparity $\hat{g}'$ using the Eq.( 10) in the main text
>
> **if** $\hat{g}' \leq 0$ **then**
> > Determine the gradient direction $d^t$ by solving the LP problem in the Eq.(12) in the main text;
>
> **end**
> **else**
> > Determine $d^t$ by solving the problem in the Eq.(13) in the main text;
>
> **end**
> Update the model $\theta^{t+1} = \theta^t + \eta d^t$;
> **if** $||d^t|| \leq \epsilon_d$ **then**
> > Decay the parameter $\delta_l^t$ and $\delta_g^t$: $\delta_l^{t+1} = \beta \cdot \delta_l^t$, $\delta_g^{t+1} = \beta \cdot \delta_g^t$
>
> **end**
> **else**
> > $\delta_l^{t+1} = \delta_l^t$, $\delta_g^{t+1} = \delta_g^t$
>
> **end**
> Until converge to **fair min-max solution** $h^1$;

**end**

**Stage 2:** continue to optimize $h^1$ to achieve Pareto optimality

**for** $t = 1, 2, ...T^2$ **do**

> *Local clients:*
> The center server sends the global model $h^t$ to all local client;
> Local clients evaluate the performance of $h^t$ and obtain the local gradients $\nabla_{\theta_t} l_i$;
> *Center server:*
> Determine $d^t$ by solving the LP problem in the Eq.(15) in the main text;
> Update the model $\theta^{t+1} = \theta^t + \eta d^t$;
> Until converge to $h^*$;

**end**

**Output:** the **fair Pareto min-max** solution $h^*$.

---

# B  Proof of Theorem 1

Firstly, $\mathcal{H}^{FP}$ and $\mathcal{H}^{FU}$ are always non-empty from their definitions. Suppose $h \in \mathcal{H}^{FU} \subset \mathcal{H}^F$. From the definitions of $\mathcal{H}^{FP}$ and $\mathcal{H}^{FU}$, we have (1) $\exists h' \in \mathcal{H}^{FP}$, s.t., $h'$ dominates $h$ that $h' \preceq h \in \mathcal{H}^{FU}$; (2) $\max(l(h)) \leq \max(l(h'))$. From (1), $\max(l(h)) \geq \max(l(h'))$ holds. Combining (2), we have $\max(l(h)) = \max(l(h'))$, so $h' \in \mathcal{H}^{FU}$ holds. Therefore, $h' \in \mathcal{H}^{FU} \cap \mathcal{H}^{FP} \neq \varnothing$ holds.

# C  Convergence Derivation of FCFL

We will prove the convergence of our method in this section. FCFL reaches a min-max Pareto fair model by a two-stage process and we aim to identify a gradient direction $d$ to optimize the model in

each iteration. For $N$ clients with $N$ corresponding objectives $[l_1, l_2, . l_N]$ and a given direction $d$, we have the following proposition, i.e.,

**Proposition 1.** *Given the gradient direction $d$ with all $d^T \nabla_\theta l_i < 0$, there exists $\eta^0$, such that:*

$$
\begin{aligned}
&l_i(h_{\theta^{t+1}}) < l_i(h_{\theta^t}), \forall i \in \{1, 2, ., N\} \\
&\theta^{t+1} = \theta^t + \eta \cdot d, \forall \eta \in [0, \eta^0].
\end{aligned} \tag{S.1}
$$

*Proof.* Consider Taylor's expansion with Peano's form of the differentiable objective function $l_i(h_{\theta^{t+1}})$:

$$
l_i(h_{\theta^{t+1}}) = l_i(h_{\theta^t + \eta \cdot d}) = l_i(h_{\theta^t}) + \eta d^T \nabla_{\theta^t} l_i + o(\eta), \tag{S.2}
$$

where $o(\eta)$ denotes a function that approaches 0 faster than $\eta$. Specifically, $\forall \epsilon > 0$, there exists $\eta^0$, such that $o(\eta) < \epsilon \eta$ for all $\eta \in [0, \eta^0]$. Now we set $\epsilon = -d^T \nabla_{\theta^t} l_i > 0$, there exists $\eta^0$, such that $l_i(h_{\theta^{t+1}}) - l_i(h_{\theta^t}) = \eta d^T \nabla_{\theta^t} l_i + o(\eta) = -\eta \epsilon + o(\eta) < 0$ for all $\eta \in [0, \eta^0]$. $\square$

We will prove the convergence of our method in two steps. First, when optimizing $h$ to achieve min-max performance with MCF as in Eq.( 11) in the main text, we optimize either $\hat{l}$ or $\hat{g}'$ and keep $\hat{l}$ without ascent in each iteration. The monotonic decreasing $\hat{l}$ means $h$ will converge to $h^1$ satisfying fairness and min-max constraints. Then, we continue to optimize $h^1$ to achieve Pareto optimality by controlling the gradient direction $d$ without causing ascent for all objectives in Eq.(15) in the main text. The objective $\frac{1}{N} \sum_{i=1}^{N} l_i$ will monotonically decrease until convergence as we constrain all objectives to descend or remain unchanged defined in Eq.(15) in the main text. Assuming all objectives and their derivatives are bounded, the formal and detailed derivations are as follows.

**Convergence of Constrained Min-Max Optimization** To prove the convergence of the Constrained Min-Max Optimization procedure, we firstly give Lemma 1 as follows:

**Lemma 1.** *In each iteration, the surrogate maximum function $\hat{l}_{max}(h_{\theta^t}, \delta_l^t)$ decreases:*

$$
\hat{l}_{max}(h_{\theta^{t+1}}, \delta_l^t) \le \hat{l}_{max}(h_{\theta^t}, \delta_l^t) \tag{S.3a}
$$

$$
\hat{l}_{max}(h_{\theta^t}, \delta_l^{t+1}) \le \hat{l}_{max}(h_{\theta^t}, \delta_l^t) \tag{S.3b}
$$

$$
\hat{l}_{max}(h_{\theta^{t+1}}, \delta_l^{t+1}) \le \hat{l}_{max}(h_{\theta^t}, \delta_l^t). \tag{S.3c}
$$

*Proof.* We prove Eq.(S.3a) in two cases: 1) $\hat{g}_{max}(h_{\theta^t}, \delta_g^t \le 0)$: we obtain the gradient direction $d$ by solving Eq.(12) in the main text. As we choose $d = -\nabla_{\theta_t} \hat{l}$, $d^T \nabla_{\theta_t} \hat{l} < 0$ holds. According to Proposition 1, as $\min d^T \nabla_{\theta_t} \hat{l} \le d^T \nabla_{\theta_t} \hat{l} < 0$, we prove $\hat{l}_{max}(h_{\theta^{t+1}}, \delta_l^t) \le \hat{l}_{max}(h_{\theta^t}, \delta_l^t)$ as in Eq.(S.3a); 2) as $\hat{g}_{max}(h_{\theta^t}, \delta_g^t > 0)$, we obtain the gradient direction $d$ by solving Eq.(13) in the main text. If we choose $-d$ which lies on the angular bisector of the angle formed by $\nabla_{\theta^t} \hat{l}$ and $\nabla_{\theta^t} \hat{g}$, we have $d^T \nabla_{\theta^t} \hat{l} \le 0$ and $d^T \nabla_{\theta^t} \hat{g} \le 0$. As the optimal solution $d^*$ of Eq.(13) in the main text satisfies $d^{*T} \nabla_{\theta^t} \hat{l} \le 0$ and $d^{*T} \nabla_{\theta^t} \hat{g} \le d^T \nabla_{\theta^t} \hat{g} \le 0$, we prove $\hat{l}_{max}(h_{\theta^{t+1}}, \delta_l^t) \le \hat{l}_{max}(h_{\theta^t}, \delta_l^t)$ as in Eq.(S.3a) in this case. While the SMF $\hat{l}_{max}(\theta^t, \delta_l^t)$ monotonically increases with respect to $\delta_l^t$, Eq.(S.3b) holds as $\theta^{t+1} = \beta \cdot \theta^t < \theta^t$. From Eq.(S.3a) and Eq.(S.3b), we have $\hat{l}_{max}(h_{\theta^{t+1}}, \delta_l^{t+1}) \le \hat{l}_{max}(h_{\theta^t}, \delta_l^{t+1}) \le \hat{l}_{max}(h_{\theta^t}, \delta_l^t)$ as in Eq.(S.3c) shows. $\square$

Combining the conclusion that $\hat{l}_{max}(h_{\theta^t}, \delta_l^t)$ decreases monotonically in each iteration in Lemma 1 with $\hat{l}_{max}(h_{\theta^t}, \delta_l^t) \ge 0$, we prove the convergence of constrained min-max optimization described in Section 4.2.

**Convergence of Constrained Pareto Optimization** Constrained Pareto optimization procedure ensures the property as follows:

**Lemma 2.** *In each iteration, the objective $l(h_{\theta^t})$ decreases or remains unchanged:*

$$l_i(h_{\theta^{t+1}}) \leq l_i(h_{\theta^t}), \; \forall i \in \{1, ., N\} \,. \tag{S.4}$$

*Proof.* We obtain the gradient direction $d$ by solving Eq.(15) in the main text during constrained Pareto optimization. If there exists a solution $d$ which satisfies all constraints in Eq.(15) in the main text, we have $d^T \nabla_{\theta^t} l_i \leq 0, \forall i$ which means all objectives will not increase in this iteration as Lemma 2 shows. If there is no solution to Eq.(15) in the main text, we achieve Pareto optimality and the algorithm converges. $\qquad\square$

# D    Time Complexity Analysis

FCFL  scales linearly with the dimension ($n$) of the model parameters. In our constrained min-max optimization procedure, the computation of $\hat{l}_m(h, \delta_l)$, $\hat{g}'_m(h, \delta_g)$ in Eq.(10) has runtime of $O(N)$. With the current best LP solver [1], the LP problem with k variables and $\Omega(k)$ constraints has runtime of $O^*(k^{2.38})$ [1]. The LP problem in Eq.(12) and Eq.(12) has 2 variables and at most 4 constraints (3 constraints for $d \in \overline{G}$ and 1 constraint for $d^T \nabla_\theta \hat{l} \leq 0$) so the runtime is $O^*(2^{2.38})$. The LP problem in Eq.(15) has $N$ variables and at most $2N + 2$ constraints so the runtime is $O^*(N^{2.38})$. In deep model in FL, usually $n \gg N$.

# E    Implementation Details and Additional Experimental Results

## E.1    Implementation Details

Following the experiment setting on Adult dataset in works [2, 3], we use the original Adult dataset and 66% samples are training samples and 33% are test samples. We also split the 33% as test sets and train models on the remaining 67% on eICU dataset. For the experiments in fairness constrained setting, we randomly split the eICU dataset with this ratio to run all experiments five times and report the average performance. We delete the sensitive attribute when training the model in fairness-constrained setting. All models are based on Logistic Regression and are trained and evaluated on randomly split datasets. While the original function for measuring $\Delta DP$ or $\Delta EO$ in Eq.(2) in the main text is not differentiable as there is indicator function $\hat{Y} = 1_{h(X) \geq 0.5}$, we use a surrogate differentiable function to approximate the indicator function $\hat{Y} = \frac{1}{1+(\frac{1-h(X)}{h(X)})^{10}}$ during training. We implement our method with Pytorch and determine all hyper-parameters (including learning rate, the decay rate of $\delta_l$, $\delta_g$, etc.) by evaluating different combinations on the training set. We run all experiments 5 times and report the average results with stds. For all baselines except FedAve+FairReg, we use the source implementation for comparison with the optimal hyper-parameters. For the baselines on fairness-constrained experiments given uniform fairness budgets, 1) if the baselines can satisfy MCF, we try to optimize the model disparities to achieve MCF with the optimal model utilities; 2) if the baselines cannot satisfy MCF after exhaustive trying, we minimize the model disparities with reasonable model utilities. For gradient computation, we use SGD optimizer for each local client. For more algorithm details, the source code of our method is available at `https://github.com/cuis15/FCFL`.

## E.2    Training Devices

We train all our models on our local Linux server with 8 GeForce RTX 2080 Ti GPUs.

## E.3    Data Asset

Adult [4] is public data that anyone can download and use it freely. eICU [5] is a dataset for which permission is required. We followed the procedure on the website `https://eicu-crd.mit.edu` and got the approval for this dataset.

---

[1] We use $O^*$ to hide $k^{o(1)}$ and $log^{O(1)}(1/\delta)$, $\delta$ being the relative accuracy. Detailed information is in [1]

88 **E.4    Additional Results on Fairness Constraint Experiment**

89 We show the statistic results on eICU in Table 1.

Table 1: the Statistic Results on LoS Prediction with MCF.

| Methods | utility mean(%) | utility min (%) | disparity max (%) |
|---|---|---|---|
| MMCF | $65.8 \pm .0$ | $61.4 \pm .0$ | $11.9 \pm .0$ |
| FA | $69.5 \pm .05$ | $60.1 \pm .09$ | $11.5 \pm 1.2$ |
| FedAve+FairReg | $67.6 \pm .9$ | $59.3 \pm 1.2$ | $11.0 \pm 4.5$ |
| ours($\epsilon = 0.1$) | $69.1 \pm 1.3$ | $\mathbf{61.4} \pm .3$ | $\mathbf{9.3} \pm 1.4$ |
| ours($\epsilon = 0.05$) | $69.1 \pm 1.5$ | $\mathbf{60.9} \pm .5$ | $\mathbf{6.8} \pm 1.1$ |

90 Table 1 shows the statistical results of LoS prediction with equal fairness budgets. Our method
91 achieves the min-max performance as we set the fairness budget $\epsilon = 0.1$. All baselines cannot reduce
92 the disparities below 0.1. When we constrain all disparities $\Delta DP_i \leq 0.05$, our method reduces the
93 disparities significantly compared to baselines with a comparable min-max accuracy $60.9\%$.

94 **Income Prediction with sensitive attribute being *gender* and equal fairness budgets**

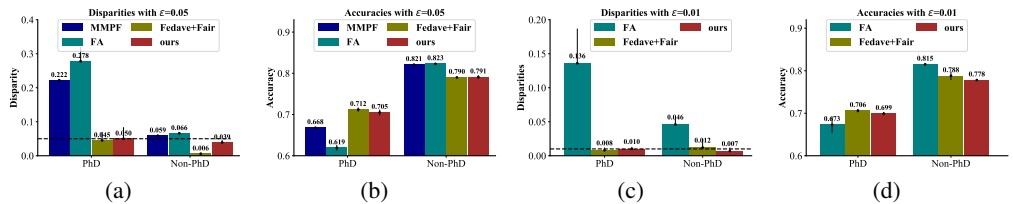

Figure 1: The disparities and accuracies on both clients as $\epsilon = 0.05$ and as $\epsilon = 0.01$ of on Adult
dataset when *gender* is the sensitive attribute.

95 In this experiment, we select *gender* as sensitive attribute. From Figure 1, with the budget $\epsilon_i =$
96 $0.05$, $\forall i \in \{1, ., N\}$, we achieve comparable min-max performance with MCF while MMPF and FA
97 violate the constraint on PhD client. As the fairness budget $\epsilon_i = 0.01$, $\forall i \in \{1, ., N\}$, all baselines
98 violate the constraint on PhD client and we maintain the utilities on both clients with MCF.

99 **Income Prediction with sensitive attribute being *gender* and client-specific fairness budgets**
100 The results of Income prediction with client-specific fairness constraints in Figure 2. The disparities
101 of both clients decrease significantly as in Figure 2(a) as $w$ decreases. With a decreasing fairness
budget, the utilities of both clients slightly decreases as in Figure 2(b).

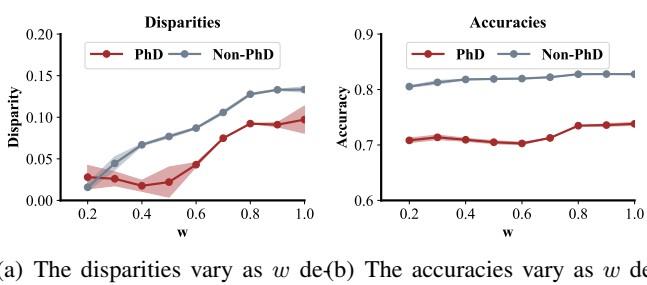

(a) The disparities vary as $w$ de-(b) The accuracies vary as $w$ de-
creases                        creases

Figure 2: Client-specific constraint experiment on Adult dataset with sensitive attribute being *gender*.

102

103 **Income prediction using EO metric** Besides DP [6] metric, we verify the effectiveness of our
104 method using another metric *Equal opportunity* (EO) [7] which measures difference of the false
105 negative rates across different groups as in Eq.(2). Here we show the results on Income prediction
106 with the sensitive attribute being *race* given uniform fairness budgets $\epsilon_i = 0.05$ on both clients in our
107 main text.

108 The original unconstrained model causes disparities on both clients: $\Delta EO_{PhD} = 0.105$ and
109 $\Delta EO_{non-PhD} = 0.183$. Our model significantly reduces the disparities with fairness budget

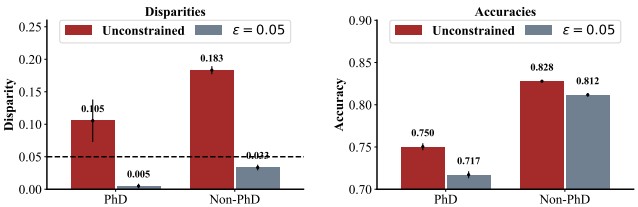

(a) the disparities of the two clients (b) the utilities of the two clients

Figure 3: The results of unconstrained optimization and fairness-constrained optimization with the disparities measured by EO and the sensitive attribute is *race*.

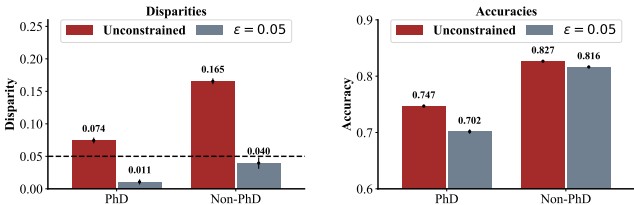

(a) the disparities of the two clients(b) the accuracies of the two clients

Figure 4: The results of unconstrained optimization and fairness-constrained optimization with the disparities measured by EO and the sensitive attribute is *gender*.

$\epsilon = 0.05$ as shown in Figure 3(a). For the model utilities on both clients shown in Figure 3(b), we achieve reasonable performances and there is a 0.03 drop on PhD client and 0.02 on non-PhD client. Similar results can also be found in Figure 4 as the sensitive attribute is *gender* in Appendix.

# F   Compatibility with Unconstrained Pareto Min-Max Optimization

## F.1   Achieving Pareto Min-Max Optimality without Fairness Constraints

Though our method is proposed for fairness-constrained multi-objective optimization, FCFL  is also compatible with unconstrained min-max optimization problems which only care about the utilities of all clients as in [2, 3]:

1. optimize $h$ to achieve min-max performance:

$$\min_{d \in \overline{G}} \quad d^T \nabla_{\theta^t} \hat{l},$$

2. optimize $h$ for Pareto optimality and min-max performance:

$$\min_{d \in \overline{G}} \frac{1}{N} \sum_{i=1}^{N} d^T \nabla_{\theta^t} l_i, \quad \text{s.t. } d^T \nabla_{\theta^t} l_i \leq 0 \quad \forall i \in \{1, ., N\},$$

where $\overline{G}$ is the convex hull of $[\nabla_{\theta^t} l_1, ., \nabla_{\theta^t} l_N]$ and we obtain the gradient direction $d$ using the gradient information of all clients without accessing the local data.

## F.2   Experiments on Improving Consistency without Fairness Constraints

We evaluate the performance of our method on the problem of improving consistency without fairness constraints described with two existing methods q-FFL [2] and AFL [3].

Our method seeks for min-max performance by optimizing the SMF $\hat{l}$ which is the upper bound of all objectives. Experimental results on Income Prediction in Table 2 show that our method achieves relatively higher performance on the worst-performing client (74.9). Besides, FCFL  achieves Pareto

Table 2: The accuracies on Income Prediction

| Methods | average (%) | PhD (%) | non-PhD (%) |
|---------|-------------|---------|-------------|
| q-FFL | $82.3 \pm .1$[2] | $74.4 \pm .9$[2] | $82.4 \pm .1$[2] |
| AFL | $82.5 \pm .5$[2] | $73.0 \pm 2.2$[2] | $82.6 \pm .5$[2] |
| ours | **82.7** $\pm.1$ | **74.9** $\pm.4$ | **82.8** $\pm.1$ |

optimality to avoid unnecessary harm to other clients. From Table 2, we maintain the performance on non-PhD client (82.8).

We conduct experiments on eICU dataset in unconstrained min-max setting. We randomly split the dataset 5 times and run the two prediction tasks. We show the statistical average results in Table 3. For LoS prediction task, we achieve higher uniformity by a higher utility on the worst-performing client compared to baselines. We also predict the in-hospital mortality as a prediction task and all methods achieve similar results.

Table 3: The accuracies on eICU without fairness constraints

| Methods | Mortality Prediction | | LoS Prediction | |
|---------|----------------------|----------------|----------------|----------------|
| | minimum (%) | average(%) | minimum (%) | average(%) |
| q-FFL | $91.7 \pm .1$ | $88.3 \pm .7$ | $57.6 \pm 2.2$ | $70.0 \pm .3$ |
| AFL | $91.7 \pm .1$ | $88.2 \pm .7$ | $58.1 \pm 2.0$ | $70.0 \pm .4$ |
| ous | $91.7 \pm .1$ | $88.3 \pm .6$ | **60.5** $\pm 2.0$ | $67.4 \pm .7$ |

# G  Discussion about Fairness Budget

## G.1  How is the fairness budget of each client relate to one another

For each fairness budegt $\epsilon_k$, it should be assigned by each client based upon its actual fairness requirements. However, the fairness budgets $\epsilon_k$ assigned by different clients are not unrelated since all fairness constraints defined by $\epsilon_k$ determine the feasible region of the model together. Due to the potential trade-off between the fairness and the utility, the fairness constraint determined by $\epsilon_k$ of the k-th client can have an impact on the utility of all clients in the federated network.

## G.2  When the fairness budget should be the same

As we stated above, the assignment of the fairness budget $\epsilon_k$ depends on the actual scenarios. In high-stake scenarios (e.g., hospitals, banks, etc.), people are highly concerned about fairness and different clients should have consistent fairness budgets. In some other scenarios such as advertising recommendations, people may be more tolerant of the difference of $\epsilon_k$ among different clients. In this case, the assignment of the fairness budget $\epsilon_k$ depends on whether it can lead to a satisfying trade-off between fairness and utility for all clients. For example, we determine the $\epsilon_k$ based upon the original disparity as the experiments presented in Sec 5.4. In addition, to obtain a budget combination that satisfies all clients, one may try different budget combinations and evaluate the performances of all clients, then all clients vote for a reasonable budget combination.

# H  Limitations and Future work

In this study, we aim to tackle the algorithmic disparity and performance inconsistency issues in federated learning. As it is hard to realize the min-max optimality and Pareto optimality by one-stage optimization, we propose a two-stage optimization framework that first achieving consistent performance then enforcing Pareto optimality. Since our framework encourages a more uniform model performance distribution, on the one hand, some clients with poor performances may be significantly improved; on the other hand, other clients whose model utility could be further improved may come to a halt.

Since we focus on addressing the local disparity, it cannot guarantee reasonable global fairness. In reality, the global disparity is also a matter of concern in a federated network. We will study how to give consideration to both local fairness and global fairness in our future work.