# OpenReview forum: "Addressing Algorithmic Disparity and Performance Inconsistency in Federated Learning"
_NeurIPS.cc/2021/Conference — NeurIPS 2021 Poster_

### Official Review · Reviewer_QSf7 · 2021-07-15

**Rating:** 7
**Confidence:** 3

**Summary:**

This paper presents a framework for finding solutions that achieve both demographic parity and performance consistency across local clients in FL. They propose first achieving min-max performance with multi-client fairness and then finding a solution that is Pareto optimal and demonstrate their method on experiments with both synthetic and real-world data.

**Limitations And Societal Impact:**

The authors have not addressed limitations and potential negative impacts. The authors should consider what types of additional disparities or unintentional concequences their local definition of fairness may introduce.

**Main Review:**

Proposing individual fairness criteria for each client is novel (as far as I am aware) and can be very important in many domains.

The proposed multi-client fairness definition can be better explained and justified. How are the $\epsilon$ related to one another? When should they be the same? What kind of disparity is introduced when $\epsilon_k$ 's are different across different k’s? Can small $\epsilon_k$ across a large k lead to large global disparity?


Does there always exist a Pareto-optimal Min-Max Performance model that satisfies MCF? It seems like the set of $\mathcal{H}^{FP}$ is defined independently from the $\mathcal{H}^{FU}$? How do we know $H^*$ the intersection is non-empty?

Not sure what line 193-194: “minimizing the current maximum value of all objectives directly in Eq(7) can cause another worse objective and can be computational hard when faced with a tremendous amount of clients” means. Is this saying runtime is exponential in k?

This submission could benefit from grammar fixes but the grammar issues did not prevent me from understanding the paper.


**Time Spent Reviewing:**

2

---

> ### Author Response · Authors · 2021-08-08
> **Response to Reviewer QSf7**
>
> We would like to thank the reviewer for acknowledging the significance of our proposed problem and the very insightful reviews. Below are our response to the comments:
>
> * **To the comment "The proposed multi-client fairness definition ...":**
>     1. **Question 1:How is the $\epsilon$ related to one another?**
>     **Answer:** we would like to explain that $\epsilon_{k}$ should be assigned by each client based upon its actual fairness requirements. However, the fairness budgets $\epsilon_{k}$ assigned by different clients are not unrelated since all fairness constraints defined by $\epsilon_{k}$ determine the feasible region of the model together. Due to the potential trade-off between the fairness and the utility, the fairness constraint determined by $\epsilon_{k}$ of the k-th client can have an impact on the utility of all clients in the federated network.
>
>     2. **Question 2: When should they be the same?**
>     **Answer:** As we stated above, the assignment of the fairness budget $\epsilon_{k}$ depends on the actual scenarios. In high-stake scenarios (e.g., hospitals, banks, etc.), people are highly concerned about fairness and different clients should have consistent fairness budgets. In some other scenarios such as advertising recommendations, people may be more tolerant of the difference of $\epsilon_{k}$ among different clients. In this case, the assignment of the fairness budget $\epsilon_{k}$ depends on whether it can lead to a satisfying trade-off between fairness and utility for all clients. For example, we determine the $\epsilon_{k}$ based upon the original disparity as the experiments presented in Sec 5.4. In addition, to obtain a budget combination that satisfies all clients, one may try different budget combinations and evaluate the performances of all clients, then all clients vote for a reasonable budget combination.
>
>     3. **Question 3: What kind of disparity is introduced when $\epsilon_{k}$'s are different across different k’s?**
>     **Answer:** Due to the data distribution discrepancy in the federated network, different clients own different Pareto Front of the two objectives (fairness and utility), which is a natural disparity among different clients. Therefore, it is necessary to allow the clients to choose the trade-off they want. Our framework achieves it by allowing the clients to assign their fairness budgets $\epsilon_{k}$'s. As the reviewer questioned, different $\epsilon_{k}$'s across different k's may induce a relatively large global disparity as the global disparity can be affected by the clients with a large $\epsilon_{k}$.
>
>     4. **Question 4: Can small $\epsilon_{k}$ across a large k lead to a large global disparity?**
>     **Answer:** For the relationship between the global disparity and the number of clients $k$, we would like to explain that the global disparity is directly related to demographic distributions of the clients rather than the number of the clients $k$ given a bounded local disparity across all clients. There are two naive examples for demonstrating it:
>
>
> **Example 1:** **a large k with a small global disparity**
>
> Suppose there are $k$ ($k$ is large) clients and all clients have N samples and the same demographic distribution ($P(A^{i} = 1) = P(A^{j} = 1) \forall i,j$) and all clients achieve low local disparities ($\Delta DP_{k} \leq 0.001$). The global disparity satisfies
> $$\begin{aligned}
> & \Delta DP = \\
> & \left|   \frac{\sum_{i=1}^{k} N \cdot P(A^{i} = 1) \cdot  P(\hat{Y^{i}} = 1 | A = 1)  }{ \sum_{i=1}^{k} N \cdot P(A^{i} = 1) } -   \frac{\sum_{i=1}^{k} N \cdot P(A^{i} = 0) \cdot  P(\hat{Y^{i}} = 1 | A = 0)  }{ \sum_{i=1}^{k} N \cdot P(A^{i} = 0) }  \right| \\
> & =  \frac{1}{k} \cdot \left|  \sum_{i = 1}^{k} P(\hat{Y^{i}}  = 1| A = 1)    -   \sum_{i = 1}^{k} P(\hat{Y^{i}} = 1 | A = 0)     \right| \\
> & \leq \frac{1}{k} \sum_{i =1}^{k} \left|  P(\hat{Y^{i}} = 1 | A = 1)  -    P(\hat{Y^{i}} = 1 | A = 0) \right| \\
> & = \frac{1}{k} \sum_{i = 1}^{k} \Delta DP_{k} \\
> & \leq 0.001
> \end{aligned}$$
>
> **Example 2:** **a small k with a large global disparity**
>
> Suppose there are 2 clients with small local disparities and the distributions are in the following tables:
>
>
>
> | **Client 1** | A = 1    | A = 0    |
> |:--------:    |:--------:|:--------:|
> |$\hat{Y}$ = 1 |   72     |   18     |
> |$\hat{Y}$ = 0 |   8      |   2      |
>
>
>
> | **Client 2** | A = 1    | A = 0    |
> |:--------:    |:--------:|:--------:|
> |$\hat{Y}$ = 1 |   2      |   8     |
> |$\hat{Y}$ = 0 |   18     |   72     |
>
>
>
> From the two tables, both clients have low local disparities as $\Delta DP_{1} = \Delta DP_{2} = 0$. However, the global disparity is $\Delta DP = 0.48$.
>
> * **To the comment "Does there always exist...":** though $\mathcal{H}^{FP}$ is defined independently from $\mathcal{H}^{FU}$ as the reviewer mentioned, there always exists one (or more) Pareto-optimal Min-Max Performance model that satisfies MCF because of the property of Pareto optimality. We can briefly prove it as follows.
> Firstly, $\mathcal{H}^{FP}$ and $\mathcal{H}^{FU}$ are always non-empty from their definitions.
> Suppose $h \in \mathcal{H}^{FU} \subset \mathcal{H}^{F}$. From the definitions of $\mathcal{H}^{FP}$ and $\mathcal{H}^{FU}$, we have
> (1) $\exists h' \in \mathcal{H}^{FP}$, s.t., $h'$ dominates $h$ that $h' \preceq h \in \mathcal{H}^{FU}$;
> (2) $\max(l(h)) \leq \max(l(h'))$.
> From (1), $\max(l(h)) \geq \max(l(h'))$ holds. Combining (2), we have $\max(l(h)) = \max(l(h'))$, so $h' \in \mathcal{H}^{FU}$ holds. Therefore, $h' \in \mathcal{H}^{FU} \cap \mathcal{H}^{FP} \neq \varnothing$ holds.
>
> * **To the comment about the contents of line 193-194,** we would like to explain that it means minimizing the maximum value of all objectives by iteratively searching for and optimizing the current largest objective without considering other objectives may be stuck in a non-convergent loop. We propose to smooth the max function considering all objectives simultaneously in Eq.(9) and prove the convergence of our method (detailed proof please refer to Appendix B). For the time complexity, our method minimizes the maximum of all objectives by solving the two LP problems (Eq.(10) and Eq.(14)) which has polynomial time complexity with respect to the number of clients $k$ and detailed analysis about the time complexity is in Appendix C.
>
> * **To the comment about the limitations and potential negative impacts,** we are grateful to the reviewer for the advice. Since we focus on addressing the local disparity, it cannot guarantee reasonable global fairness as mentioned in Example 2. In reality, the global disparity is also a matter of concern in a federated network. We will study how to give consideration to both local fairness and global fairness in our future work.

---

> ### Comment · Reviewer_QSf7 · 2021-08-20
> **Thanks for the clarification**
>
> Hi Authors,
>
> Thanks for answering my questions and addressing my concerns.
> Happy to change my decision to accept.

---

### Official Review · Reviewer_Nsjv · 2021-07-16

**Rating:** 6
**Confidence:** 3

**Summary:**

The paper studies how to solve the model fairness and performance consistency among clients in federated learning. There exist work on model fairness and performance inconsistency separately as mentioned in the paper and the contribution of this work is to provide a FL framework which considers both effects. The problem is formulated as a constrained min-max pareto optimization problem. The authors use a two-stage optimization method to approximate the optimal solution.

**Limitations And Societal Impact:**

Yes, the authors have adequately addressed the limitations and potential negative societal impact of their work?

**Main Review:**

The paper brings local model fairness into federated learning that allows clients to set its own “fairness budget”. The model fairness is an important problem in recent machine learning communities to avoid the model being biased by the sensitive attributes. The problem is quite interesting and practical. It is not evident to see how to combine fairness with performance consistency in federated learning. The authors gave a solution easy to follow and performs well in logistic regression problems.  The paper is well organized and the ideas are easy to follow although it makes use of the existing techniques.

I have the following comments:
1. The main algorithm 1 in Appendix for FCFL is presented in a centralized way. It would be better to present it from both the client and the server perspectives.  It is not clear from the main paper how to parallelize the calculations of FCFL.
2. The experiments part is a little bit weak as only logistic regression problems are considered. It would be better to show the neural network performance.
3. It seems the definition of $\triangle DP_k$ in Eq. 2 is for binary classification. Just for curiosity, what does the  $\triangle DP_k$ look like for multiple classification problems?


**Time Spent Reviewing:**

5

---

> ### Author Response · Authors · 2021-08-09
> **Response to Reviewer Nsjv**
>
> We would like to thank the reviewer for the positive and valuable comments. Below are our response to the comments:
>
> * **To the comment "It is not evident to see how to combine fairness with performance consistency...,"** we would like to explain that enforcing the model to be fair may exacerbate the performance inconsistency due to the potential trade-off between the fairness and performance. For example, it may result in worse performance on the worst-performing client when we set a stricter fairness constraint as shown in Figure 5. The learned model with significantly poor performance would not be approved by the worst-performing client in reality. To enhance the usability of the learned model for all clients, we propose to improve the bottom line of the performances (performance consistency) while addressing the local disparity on all clients.
>
> * **To the comment about the algorithm 1 in Appendix,** we would like to thank the reviewer for the advice and will redisplay the algorithm following the suggestion.
>
> * **To the comment about the experiments using logistic regression,** we would like to explain that logistic regression usually performs better than other models on the benchmark datasets used for the fairness research. Because the learning tasks usually have relatively low feature space dimensions, a model with a stronger feature extractor may cause severe overfitting. Therefore, existing related works mainly use logistic regression model (e.g., [4,5,6,21]) as the base model. We follow the suggestion and conduct the experiments using a two-layer neural network as the base model and the results are as follows:
>
> **Adult (fairness metric: DP, sensitive attribute: Race, fairness budget: 0.05)**
>
>
> | methods     | Acc@Client 1 | Acc@Client 2 | Dis@Client 1 | Dis@Client 2 |
> |:-------:    |:--------:|:--------:|:--------:|:--------:|
> |  MMPF       |   0.707   |   0.825   |   0.225   |   0.083   |
> |  FA         |   0.663   |   0.822   |   0.025   |   0.062   |
> | FedAve+Fair |   0.722   |   0.792   |   0.059   |   0.004   |
> |  ours       |   0.729   |   0.819   |   0.015   |   0.047   |
>
>
> **eICU (fairness metric: DP, sensitive attribute: Race, fairness budget: 0.10)**
>
>
>
> | methods     | Min@Acc | Max@Dis | Acc@Client 1 | Acc@Client 2 | Acc@Client 3 | Acc@Client 4 |
> |:-------:    |:--------:|:--------:|:--------:|:--------:|:-------:    |:--------:|
> |  MMPF       |   0.576   |   0.100   |   0.685   |0.639   | 0.623  |   0.602   |   0.634   |
> |  FA         |   0.596   |   0.060   |   0.802   |   0.715   |   0.710   |   0.650   |
> | FedAve+Fair |   0.618   |   0.061   |   0.801   |   0.717   |   0.712   |   0.639   |
> |  ours       |   0.630   |   0.058   |   0.790   |   0.707   |   0.701   |   0.634   |
>
>
>
>
> | methods     | Acc@Client 5 | Acc@Client 6 | Acc@Client 7 | Acc@Client 8 | Acc@Client 9 | Acc@Client 10 | Acc@Client 11 |
> |:-------:    |:--------:|:--------:|:--------:|:--------:|:-------:    |:--------:|:--------:|
> |  MMPF       |   0.634   |   0.683   |   0.630   |   0.649   |   0.580   |   0.576   |   0.586   |
> |  FA         |   0.706   |   0.821   |   0.688   |   0.758   |   0.633   |   0.596   | 0.642  |
> | FedAve+Fair |   0.728   |   0.815   |   0.687   |   0.754   |   0.628   |   0.619   |   0.664   |
> |  ours       |   0.720   |   0.758   |   0.695   |   0.744   |   0.632   | 0.630  | 0.665  |
>
>
>
> From the results shown in the above tables, our method achieves higher performances on the worst-performing clients while satisfies fairness requirements.
>
> * **To the comment about the fairness metric when faced with multiple classification problems,** we would like to share our views on this topic as follows. The core notion adopted by algorithmic fairness is envy-freeness, which mandates that no subgroup should envy another subgroup. The algorithm prediction should be independent of the demography. According to this principle, any metric which is used for measuring the distance between the distributions of the predicted results of the two subgroups ($P(\hat{Y} | A = 0)$ and $P(\hat{Y} | A = 1)$) can be a disparity definition. For example, we use infinite norm to define the local disparity $\Delta DP_{k}$ in a multiple classification problem:
> $\Delta DP_{k} = \max_{i} ( \left| P(\hat{Y^{k}}  = i | A^{k} = 1)  - P(\hat{Y^{k}}  = i | A^{k} = 0)     \right| )$
> It is worth mentioning that the definition of algorithmic fairness is determined by the demand of practice. In a multiple classification problem, people may need to design the distance metric based on the preference of the predicted results to satisfy envy-freeness.

---

### Official Review · Reviewer_qPfz · 2021-07-16

**Rating:** 6
**Confidence:** 4

**Summary:**

This paper first introduces a new task about federated learning in which fairness and consistency are jointly considered, then proposes a solution named FCFL to solve the problem. After introducing fairness, the problem is transferred to a min-max optimization problem, and the two-stage optimization algorithm works well on it. The main contributions are as follows:

1) Introduce a new task which jointly considers model fairness and consistency in federated learning,
2) Propose an optimization method called FCFL to address the problem,
3) Conduct experiments on a synthetic dataset and two public datasets and compare the proposed algorithm with three baseline methods, and show that the proposed method outperforms the rest.

**Limitations And Societal Impact:**

The authors has discussed the limitations in the appendix. However, potential negative social impact is not discussed. Moreover, more limitations should be considered, for example, from the aspect of how important fairness is in federated learning, as it has never be mentioned before.

**Main Review:**

Both method and task in this paper are new, and the proposed method is a good utilize of several existing approaches. Related works are cited in a proper way. The idea of the task is interesting, as few work has ever considered fairness in a federated learning setting and introducing it makes the whole problem a min-max optimization. Most of the paper is well-written, a few details can be further improved (e.g. introduce full name of FCFL when it first appears).The paper is technically sound, both theoretical analysis and experimental results are sufficient. However, I believe more models in the area of federated learning should be taken into consideration. Some recent works are focusing on the inconsistency caused by non-iid in federated learning and proposed good solutions (e.g.  FedNova). Weakness of the work should also be discussed, especially the possible usage of this framework. The results are reproducible, with some help of the authors. The experimental results are nice, researchers should be interested to further build on them. The datasets are existed. Since this is a brand new task, no previous methods can be used to compare with, but the results still show that performance of the framework is generally good.

**Time Spent Reviewing:**

4

---

> ### Author Response · Authors · 2021-08-08
> **Response to Reviewer qPfz**
>
> We would like to thank the reviewer for recognizing the contributions of our work and providing valuable feedback. Below are our response to the comments:
>
> * **To the comment "a few details can be further improved."**, we are grateful to the reviewer for the kind suggestion and we will revise our paper carefully following the suggestion.
>
> * **To the comment "... more models ... should be taken into consideration",**, as the reviewer stated, no previous methods can be used to compare with as we study a new problem. FedNova focuses on addressing the objective inconsistency in federated optimization by providing a normalized averaging method. We thank the reviewer for the suggestion and more recent works will be discussed in the related work.
>
> * **To the comment about the limitations and societal impact,**, we would like to thank the reviewer for the guidance on the limitations. As the reviewer stated, fairness in federated learning is an important concern. Similar to prior work on the fairness issues in non-FL settings, the fairness achieved by the learned model may be vulnerable to adversarial attacks and our work is no exception. As our work focuses on local fairness in federated learning, global fairness is also a matter of concern as Review QSf7 mentioned. For these problems, we expect to draw researchers further exploration.

---

### Official Review · Reviewer_qMLa · 2021-07-17

**Rating:** 6
**Confidence:** 4

**Summary:**

This paper proposes a federated learning algorithm which is fair and gives consistent performance across different clients in the following sense.  To achieve fairness, some measure of disparity is considered and enforced as a constraint for each client (i.e. each client has an upper bound on how high the disparity measure can go).  To achieve consistency, the objective is to find model parameters that minimize the maximum loss incurred across the clients.  This contrasts with prior work which does not directly address the fairness locally at each client.  Compared to [21], this paper adds constraints on a disparity measure for each client, in addition to the min-max objective.  Compared to [6], this paper considers a fairness constraint at each client as opposed to a single, global fairness constraint.

The proposed gradient descent-based algorithm first aims to satisfy the fairness constraints, and then it aims to find a Pareto optimal solution (w.r.t the clients’ utilities) while continuing to satisfy the fairness constraints.  The proposed algorithm replaces the (non-smooth) max’s in the problem formulation with a smooth surrogate, which is basically the log-sum-exp trick.

The proposed method is evaluated on three datasets with respect to prediction accuracy as the loss/utility metric and two different fairness metrics: demographic parity and equal opportunity.  The results show that the proposed method typically performs significantly better with respect to the fairness measure, with a small loss in accuracy compared to other methods which do not achieve the same level of fairness.


**Limitations And Societal Impact:**

Some discussion about the choice of fairness metric may be useful (this paper looks at demographic parity and equal opportunity).  I do not see a major potential for negative societal impact directly stemming from this work.

**Main Review:**

This paper considers two currently relevant topics - fairness in ML and federated learning.  The main contribution here is the problem formulation and the experimental results, which are interesting.  I would not consider the algorithm design (including the tricks such as replacing max with log-sum-exp) very novel.  Below I outline some issues I have with the current presentation.  Overall, I would be okay with accepting this paper.

 - The authors claim that their method converges in theory, but no claim is made as to the convergence rate.
 - The experiments seem to focus on reporting the accuracy and fairness performance.  The proposed method involves solving an LP at each iteration (for both stages of the algorithm).  What is the impact on the runtime of solving these?  How does the overall runtime of the proposed method compare to the baselines considered?  Are the improvements to fairness/accuracy coming at a significant computational cost compared to other methods?
 - The number of clients considered in the experiments seems rather small.  How does the proposed method perform in situations where there are significantly more clients?

Minor comments:
 - Line 33 - “in FL scenario” -> “in a FL scenario”
 - Line 42 - “There have been researches” -> “There has been prior work”
 - Line 170 - “in deep model” -> “in a deep model”



**Time Spent Reviewing:**

4

---

> ### Author Response · Authors · 2021-08-09
> **Response to Reviewer qMLa**
>
> We would like to thank the reviewer for acknowledging the highlights of our work and valuable comment. Below are our response to the comments:
>
> * **To the comment about the convergence of our proposed framework,** we would like to explain that as shown in the convergence derivation in Appendix B, we prove the convergence of our method from a constrained gradient descent perspective. The convergence rate of our model is consistent with that of the general deep learning model. We also verify this experimentally that learning convergent models of the baselines requires similar iteration times as our method. The runtime results of our methods with baselines are shown in the following table and the rigorous analysis of the convergence rate will be explored in our subsequent research.
>
>
>
> * **To the comment about the impact of LP on the runtime of our method,**
>     1. **Question 1: What is the impact on the runtime of solving these?**
>     **Answer:** As stated in Appendix C, the time complexity of the LP problems with $k$ variables is $O^{*}(k^{2.38})$. When solving the LP problems in our method, the number of variables equals the number of clients which is much smaller than the number of model parameters.
>
>     2. **Question 2: How does the overall runtime of the proposed method compare to the baselines considered?**
>     **Answer:** We compare the runtime of our method to the baselines on the two datasets and the results are shown in the following table. From the table, our method has a comparable runtime cost compared to baselines on our conducted experiments.
>
>     3. **Question 3: Are the improvements to fairness/accuracy coming at a significant computational cost compared to other methods?**
>     **Answer:** MMPF needs to optimize the weights of different subgroups in each iteration. FA requires to determine the weights of all samples in each iteration by kernel function parametrization. Compared to baselines, our method improves the utility while keeps a low disparity on all clients without coming at a significant computational cost.
>
> | methods     | Adult         | eICU        |
> | --------    | --------      | --------    |
> | MMPF        | 20 min 52 s   | 10 min 26 s |
> | FedAve+Fair | 14 min 18 s   | 13 min 59 s |
> | FA          | 16 min 22 s   | 43 min 54 s |
> | our         | 18 min 8 s    | 29 min 12 s |
>
>
>
>
> * **To the comment "... there are significantly more clients",** as we could not find such a real federated dataset with fairness issues, we would like to explain it from the two aspects:
>
>     1). **fairness/utility:** when faced with significantly more clients, our method can ensure the fairness of the learned model as we enforce it satisfies the fairness constraints on all clients as we stated in Sec 4.2. Our method achieves a constrained Pareto min-max model. Compared to FedAve, our method targets a specific model with the fairness constraints. More clients mean more constraints on the hypothesis set. Due to the potential trade-off between fairness and utility, our learned model focuses on addressing the disparity on all clients may cause degradation on the utility because of a smaller feasible region of the hypothesis set.
>
>     2). **runtime cost:** as we stated above, our method has polynomial time complexity with respect to the number of the clients. The time consumption of our methods is acceptable when faced with more clients.
>
>
> * **To the comment in Minor comments,** we thank the reviewer for pointing out the writing mistakes and will revise our paper carefully.
>
> * **To the comment about the limitations and societal impact,** we would like to thank the reviewer for providing the advice. We think that the choice of the fairness metric may depend on the actual needs of the clients in practice. For example, hospitals may need to ensure that all subgroups should have the same chance to have access to the treatment (demographic parity). Banks may be more concerned about whether people of all subgroups who are solvent have the same chance to have access to the loans (equal opportunity). This inspires us to design the algorithm that is compatible with various metrics that may be used in reality.

---

### Decision · Program_Chairs · 2021-09-27

**Decision:**

Accept (Poster)

**Comment:**

This paper studies the notion of fairness concerns in Federated Learning settings where a vanilla global model may be particularly bad on some clients versus others. Their solve this issue by maximizin the performance of the model on the worst client, leading to minimax style of fairness gurantees. The authors complement this analysis with an empirical evaluation of their method.